# mTOR Inhibitors Modulate the Biological Nature of TGF-β2-Treated or -Untreated Human Trabecular Meshwork Cells in Different Manners

**DOI:** 10.3390/biomedicines12112604

**Published:** 2024-11-14

**Authors:** Megumi Watanabe, Tatsuya Sato, Toshiyuki Yano, Megumi Higashide, Toshifumi Ogawa, Nami Nishikiori, Masato Furuhashi, Hiroshi Ohguro

**Affiliations:** 1Departments of Ophthalmology, School of Medicine, Sapporo Medical University, S1W17, Sapporo 060-8556, Japan; watanabe@sapmed.ac.jp (M.W.); megumi.h@sapmed.ac.jp (M.H.); nami076@yahoo.co.jp (N.N.); 2Departments of Cardiovascular, Renal and Metabolic Medicine, Sapporo Medical University, S1W17, Sapporo 060-8556, Japan; satatsu.bear@gmail.com (T.S.); oltomwaits55@gmail.com (T.Y.); a08m024@yahoo.co.jp (T.O.); furuhasi@sapmed.ac.jp (M.F.); 3Departments of Cellular Physiology and Signal Transduction, Sapporo Medical University, S1W17, Sapporo 060-8556, Japan

**Keywords:** TGF-β2, human trabecular meshwork, 3D culture, rapamycin, mTOR, autophagy

## Abstract

**Background/Objectives:** Mammalian target of rapamycin (mTOR) inhibition may have been suggested to have a beneficial effect on the glaucomatous human trabecular meshwork (HTM). To study the effects of the mTOR inhibitors rapamycin (Rapa) and Torin1 on the glaucomatous HTM, transforming growth factor-β2 (TGF-β2)-treated two-dimensionally (2D) and three-dimensionally (3D) cultured HTM cells were used. **Methods:** We evaluated (1) the levels of autophagy via Western blot analysis using a specific antibody against microtubule-associated protein 1 light chain 3 (LC3), (2) barrier capacity based on transepithelial electrical resistance (TEER) and fluorescein isothiocyanate (FITC) permeability (2D), (3) cellular metabolic functions (2D), (4) the size and stiffness of spheroids, and (5) the mRNA expression of ECM proteins. **Results:** TGF-β2-induced inhibition of autophagy was significantly inhibited by Rapa and Torin1. Rapa and Torin1 substantially decreased barrier capacity in both TGF-β2-untreated and TGF-β2-treated HTM cells. Cellular metabolic analysis indicated that Rapa, but not Torin1, substantially enhanced both mitochondrial and glycolytic functions of TGF-β2-untreated HTM cells. In the physical properties of spheroids, TGF-β2 resulted in the formation of down-sized and stiffened spheroids. mTOR inhibitors decreased the size but not the stiffness of TGF-β2-untreated spheroids and significantly reduced the TGF-β2-related increase in the stiffness but not the size of spheroids. The diverse effects of mTOR inhibitors on TGF-β2-untreated and TGF-β2-treated spheroids were also observed in the mRNA expression of extracellular matrix proteins. **Conclusions:** The results taken together suggest that mTOR inhibitors significantly influence the biological aspects of both a single layer and multiple layers of the TGF-β2-treated HTM and untreated HTM.

## 1. Introduction

Increased levels of intraocular pressure (IOP) induced by an increase in mechanical resistance within the trabecular meshwork (TM) during aqueous humor (AH) outflow are primarily involved in the pathogenesis of glaucomatous optic neuropathy (GON) [1,2]. Various growth factors and cytokines including transforming growth factor-beta2 (TGF-β2) have been shown to be related to the underlying mechanism [3,4,5]. AH levels of TGF-β2 (1.94 to 3.46 ng/mL) in patients with primary open-angle glaucoma (POAG) were shown to be significantly higher than those in non-glaucoma patients (0.41 to 2.24 ng/mL) [6,7], suggesting that TGF-β2 indeed induces excessive deposits of ECM within the TM, leading to elevation in POAG [8,9,10] as the glaucomatous TM phenotype [11].

Rapamycin (Rapa) was discovered in Streptomyces hygroscopicus and was later recognized to have immunosuppressive and antiproliferative effects in mammalian cells [12]. The effects of Rapa were shown to be induced by potent suppression of S6 kinase 1 (S6K1) [13,14,15] and the induction of PI3 kinase (PI3K) signaling [16]. The target of Rapa (TOR) was simultaneously discovered in yeast cells in addition to animal cells [17,18]. There is a functional complex of Rapa in mammalian cells called mTOR complex (mTORC), and two functionally distinct complexes, mTORC1 and mTORC2, were identified [17,19], despite the fact that both mTORC1 and mTORC2 are inhibited by Rapa in some cases of long-term exposure [17]. Physiologically, mTORC1 functions to modulate various biological signaling pathways related to energy metabolism and the synthesis of proteins and lipids in addition to autophagy [12]. On the other hand, dysregulated mTORC1 has been shown to induce various disorders and diseases including several genetic disorders, malignancies, and age-related diseases [20]. In contrast with mTORC1, although the pathophysiological roles of mTORC2 have not been extensively studied compared to those of mTORC1, mTORC2 has been suggested to be involved in cell survival mechanisms by regulating actin-cytoskeleton organization via activation of Akt and SGK1 [17]. In ocular pathophysiology, it has been shown that Rapa could (1) improve retinal ganglion cell survival in a rat model of chronic ocular hypertension [21], (2) markedly enhance autophagy in a monkey model of chronic hypertension [22], (3) increase AH outflow by modulating the autophagy homeostasis of TM cells [23], (4) downregulate fibrogenic changes in TGF-β2-treated HTM cells [24], and (5) dysregulate the autophagic pathway in cells isolated from the glaucomatous TM [25]. These collective findings suggest that mTOR inhibition may have a beneficial effect on the glaucomatous HTM. However, since the HTM is a complex three-dimensional (3D) architecture, additional studies using suitable in vitro models replicating the complex HTM architecture will be required for a better understanding of the effects of mTOR inhibitors on the glaucomatous HTM.

Recently, we independently developed an in vitro model suitably replicating such a 3D HTM environment, that is, multiple sheet layers, using a spheroid culture of HTM cells [26]. In this study, to elucidate the undermined mTOR inhibitory effects on the glaucomatous HTM, the effects of the mTORC1 inhibitor Rapa and the mTORC1 and mTORC2 inhibitor Torin1 on the glaucomatous HTM were studied using our newly developed in vitro 2D and 3D HTM cell culture models.

## 2. Materials and Methods

### 2.1. Two-Dimensionally (2D) and Three-Dimensionally (3D) Cultured Human Trabecular Meshwork (HTM) Cells

All studies were performed according to the tenets of the Declaration of Helsinki and were approved by the internal review board of Sapporo Medical University. As in vitro glaucomatous HTM models, immortalized HTM cells (Applied Biological Materials Inc., Richmond, BC, Canada) [27] were 2D and 3D cultured in the absence and presence of 5 ng/mL of TGF-β2 as reported previously [28]. Briefly, the HTM cells were maintained in 150 mm planar culture dishes at 37 °C in high glucose Dulbecco’s Modified Eagle Medium (HG-DMEM) medium containing 10% fetal bovine serum (FBS), 1% L-glutamine, and 1% antibiotic–antimycotic, until reaching 90% confluence by changing the medium every other day. For spheroid generation, 2D cultured HTM cells were collected and resuspended in the same culture medium supplemented with 0.25% methylcellulose. Cell numbers were adjusted to approximately 20,000 cells in 28 μL of culture medium and placed in each well of a hanging droplet 3D culture plate (# HDP1385, Sigma-Aldrich. St. Louis, MO, USA). Thereafter, spheroid culture was carried out for a 6-day period with a daily change of half of the medium with a fresh medium.

To evaluate the drug-induced effects of mTOR inhibitors on 5 ng/mL TGF-β2-treated or untreated 2D and HTM spheroids, 100 nM rapamycin (Rapa) or Torin1 was administered from Day 1 to Day 6 of the culture period. The concentrations of TGF-β2 [28,29] and mTOR inhibitors were confirmed to be the optimum concentrations based on previously reported data [30,31].

### 2.2. Transepithelial Electrical Resistance (TEER) and Fluorescein Isothiocyanate FITC Dextran Permeability Measurements of the 2D Monolayer of HTM Cells

To estimate the barrier function in the 2D HTM Monolayers, 5 ng/mL TGF-β2-untreated or TGF-β2-treated 2D HTM monolayers were subjected to TEER and FITC measurements in the absence or presence of 100 nM of an mTOR inhibitor (Rapa or Torin1) as described in a previous report [32].

### 2.3. Measurement of Real-Time Cellular Metabolic Function Using a Seahorse Bioanalyzer

To estimate the cellular metabolic functions of 2D cultured HTM cells in the absence or presence of 5 ng/mL TGF-β2 and/or 100 nM of an mTOR inhibitor (Rapa or Torin1), oxygen consumption rate (OCR) and extracellular acidification rate (ECAR) were measured using a Seahorse XFe96 Bioanalyzer (Agilent Technologies, Santa Clara, CA, USA) as described recently [33].

### 2.4. Measurements of the Size and Stiffness of HTM Spheroids

As described previously [28], the sizes of spheroids were measured under an inverted microscope (Nikon ECLIPSE TS2; Tokyo, Japan). To obtain an index, the ratio of force/displacement (μN/μm) expressing stiffness, the force required for the diameter of a single living spheroid to reach 50% of its length (μm) was measured by applying a micro-compressor system (MicroSquisher, CellScale, Waterloo, ON, Canada).

### 2.5. Western Blotting

Protein concentration was determined using the Bradford assay. Equal amounts of proteins were electrophoresed on 7.5% or 12.5% polyacrylamide gels and then blotted onto PVDF membranes (Millipore, Bedford, MA, USA). After blocking had been performed with a TBS-T buffer containing 5% nonfat dry milk or 5% BSA, the blots were incubated with 1:1000 dilutions of anti-human LC3 rabbit polyclonal antibody (Cell Signaling Technology, Beverly, MA, USA) or 1:1000 dilutions of anti-human α-tubulin mouse monoclonal antibody (Sigma Aldrich, St. Louis, MO, USA) at 37 °C for 2 h. Then, they were incubated with an HRP-conjugated secondary antibody, goat anti-rabbit antibody, or goat anti-mouse antibody (Biorad Laboratories Inc., Hercules, CA, USA) at 37 °C for 1 h after washing three times with TBS-T buffer for 10 min each time. Immunolabeled proteins were visualized by using an ECL Western blotting kit (Thermo Scientific, Rockford, IL, USA).

### 2.6. Other Analytical Methods

For qPCR analysis, total RNA extraction using an RNeasy mini kit (Qiagen, Valencia, CA, USA) and reverse transcription usng a SuperScript IV kit (Invitrogen, Waltham, MA, USA) was performed using predesigned primers and probes (Appendix A) and a StepOnePlus machine (Applied Biosystems/Thermo Fisher Scientific, Waltham, MA, USA) as previously reported [34]. All experimental data were shown by the arithmetic mean ± the standard error of the mean (SEM) and statistical analyses were performed as described in our previous report [34].

## 3. Results

To elucidate the effects of the mTOR inhibitors on the glaucomatous HTM, a recently established in vitro glaucomatous HTM model [35] using TGF-β2-treated 2D and 3D cultured HTM cells, which replicate a single layer and multiple layer structures of the HTM, respectively, was used. At first, to evaluate the effects of mTOR inhibitors on the levels of autophagy of 2D HTM cells, protein levels of LC3-I and LC3-II were determined via Western blot analysis. As shown in Figure 1, upon administering TGF-β2, immunoreactivity against LC3-II, but not that against LC3-I, was significantly reduced, suggesting that levels of autophagy were reduced by TGF-β2. Although the levels of both LC3-I and LC3-II of the untreated TGF-β2 HTM cells were not altered by Rapa or Torin 1, levels of LC3-II, not those of LC3-I, of TGF-β2-treated HTM cells were substantially increased by mTOR inhibitors. TEER and FITC dextran permeability measurements of a 2D HTM monolayer (Figure 2) showed that the administration of 5 ng/mL TGF-β2 caused a significant increase in TEER values and a significant decrease in FITC dextran permeability, as was observed in our previous study [28]. In contrast, Rapa and Torin1 both induced a substantial decrease in TEER value and a substantial increase in FITC dextran permeability in both TGF-α2-untreated HTM cells and TGF-β2-treated HTM cells. Seahorse cellular metabolic analysis (Figure 3) showed that mono-treatment with TGF-β2 induced a metabolic shift from oxidative phosphorylation to glycolysis. In contrast, Rapa substantially increased both mitochondrial respiration and glycolysis despite no significant effects of Torin1 in TGF-β2-untreated HTM cells. However, such mTOR inhibitor-induced modulations in some metabolic indices were diminished in TGF-β2-treated HTM cells. The collective findings indicate that Rapa or Torin1 had different effects on TGF-β2-untreated 2D HTM cells and TGF-β2-treated 2D HTM cells despite having similar effects on barrier function. That is, (1) increased levels of autophagy and (2) metabolic shift from oxidative phosphorylation to glycolysis were observed during treatment with TGF-β2.

To study this issue further, the effects of mTOR inhibitors on the physical properties, size, and stiffness of HTM spheroids were evaluated. As shown in Figure 4, the sizes were reduced during the 6-day culture period, but the rate of reduction was higher in HTM spheroids treated with TGF-β2 than in the untreated control spheroids. An increase in the stiffness of HTM spheroids was induced by TGF-β2, as was observed in our previous study [28]. Mono-treatment with Rapa or Torin1 decreased the size but not the stiffness of HTM spheroids, and both mTOR inhibitors significantly reduced the TGF-β2-induced increase in stiffness but not the size of the spheroids (Figure 5). To obtain further insights into the effects of the mTOR inhibitors, the mRNA expression of extracellular matrix proteins including collagen1 (COL1), COL4, COL6, fibronectin (FN), and α smooth muscle actin (αSMA) was studied, and it was found that mTOR inhibitors induced (1) marked downregulation of the gene expression of *COL1*, *COL4*, *COL6*, and *FN* in TGF-β2-untreated HTM spheroids, and those effects were less in the TGF-β2-treated HTM spheroids, and (2) upregulation of αSMA expression in TGF-β2-treated HTM spheroids (Figure 6).

Taken together, the results suggest that the biological aspects of both a single layer and multiple layers were greatly affected by the mTOR inhibitors.

## 4. Discussion

It has been shown that mTOR, a serine/threonine kinase, is a master regulator of cellular metabolism processes including cell growth and proliferation, thereby regulating cell size and the number of cells [17,36,37], and that dysregulation of its signaling pathway is involved in the pathogenesis of various human diseases [38,39]. In addition to the regulation of cell size and the number of cells, mTOR plays a pivotal role in the regulation of autophagy [40,41,42]. These collective findings allowed us to speculate that the modulation of mTOR signaling may change cell sizes and the number of cells within the HTM, thereby affecting IOP levels via alteration of drainage of the AH outflow pathway. As we expected, in the present study, we found that mTOR inhibitors caused a marked decrease in barrier function, as shown by TEER and FITC permeability measurements in both TGF-β2-treated and TGF-β2-untreated 2D HTM monolayers (Figure 2). In addition, the stiffness of both TGF-β2-treated and TGF-β2-untreated 3D HTM spheroids was also substantially decreased, although 3D HTM spheroids were downsized with the treatment of TGF-β2 and mTOR inhibitors and their effects were not different (Figure 4). Since the TGF-β2-induced downsizing of 3D HTM spheroids was considered to be caused by an increase in fibrogenesis but not related to cell size and the number of cells as described in our previous report [35], we speculated that the mTOR inhibitor-induced decrease in the stiffness of 3D HTM spheroids was caused by a decrease in cell size and the number of cells being contained in a 3D HTM spheroid but not by fibrogenesis. In support of this speculation, mRNA expression of most of the ECM proteins was significantly downregulated by mTOR inhibitors (Figure 6). Similar to our results, Igarashi et al. reported that the TGF-β2-induced upregulation in the expression of fibronectin, COL1A1, and αSMA in HTM cells was significantly suppressed by mTOR inhibitors and that the TGF-β2-induced reduction in the migration rates of HTM cells was also inhibited by mTOR inhibitors [24]. In addition, a previous study using a rat model with elevated IOP showed that Rap caused IOP reduction through an increase of the AH outflow facility, presumably via regulation of the RhoA-related actin cytoskeleton in TM cells, and also protected retinal ganglion cells by inhibiting the activation of glial cells and the release of proinflammatory factors, suggesting that targeting Rap may be an effective strategy for the treatment of glaucoma [43].

It is known that autophagy breaks down macromolecules and organelles in lysates through lysosomal enzymes [44], and several biochemical pathways including lipidation of the autophagosome marker LC3-I into LC3-II have been shown to be associated with this mechanism [45]. In general, although autophagy essentially occurs in all cell types for the maintenance of cellular homeostasis, this can be promptly enhanced in response to several types of stress such as reactive oxygen species (ROS) [44,46,47]. Thus, dysregulated autophagy is known to be involved in the pathogenesis of human disorders such as age-related diseases and neurodegenerative disorders [48,49,50]. In ocular diseases, dysregulated autophagy has been shown in models of glaucomatous HTM cells caused by TGF-β2 [51], dexamethasone [52], and mechano-stretch [53]. A recent study in which transcriptome and functional analyses were conducted by using primary cultured TM cells with Atg5/7-deficiency indicated alterations in the gene expression of various fibrotic genes including TGF-β2 [51]. In addition, it was shown that genetic and pharmacological suppression of autophagy attenuated TGF-β-induced fibrosis, suggesting crosslinking of autophagy to TGF-β signaling in TM cells [51]. In addition, several studies have suggested that stimulation of autophagy in TM cells may be an early event of elevation of IOP and chronic oxidative stress [54,55,56]. Furthermore, it was shown that autophagy facilitated the degradation of a gene related to POAG myocilin [57]. In the present study, we found that mTOR inhibitors substantially increased levels of autophagy in TGF-β2-treated HTM cells but not in TGF-β2-untreated HTM cells, suggesting that mTOR inhibitors had beneficial effects on the permeability of the TGF-β2-treated HTM cells in addition to the regulation of cell size and the number of cells. Furthermore, the cellular metabolism of TGF-β2-untreated HTM cells was enhanced by mTOR inhibitors with a shift from oxidative phosphorylation to glycolysis (Figure 3) as observed in a previous study using human cardiac fibroblasts [58]. However, such beneficial effects on cellular metabolic function by mTOR inhibitors were not observed in TGF-β2-treated HTM cells (Figure 3).

In conclusion, taken together with previous observations that mTOR inhibitors ameliorated TGF-β2-induced fibrogenesis [59,60], our collective results suggest that mTOR inhibitors may have beneficial effects on (1) the cell size and number of cells and (2) the autophagy of TGF-β2-untreated (healthy) HTM cells and TGF-β2-treated (glaucomatous) HTM cells, and (3) the cellular metabolic functions of TGF-β2-untreated (healthy) HTM cells. However, mTOR is involved in various biological functions, and mTOR inhibitors thus have many pharmacological effects, including the regulation of autophagy. In addition, the effects of short-term exposure and long-term exposure to Rap have been shown to be different [12]. Therefore, additional investigations using various effectors related to mTOR signaling with primary cultured glaucomatous HTM cells will be required for a better understanding of the effects of mTOR inhibitors on glaucomatous and non-glaucomatous HTM cells.

## Figures and Tables

**Figure 1 biomedicines-12-02604-f001:**
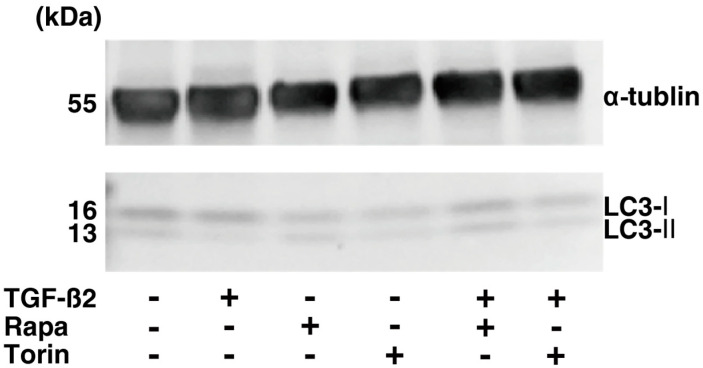
Western blotting of 2D HTM cells. Two-dimensional HTM cells treated or not treated with 5 ng/mL TGF-β2 and/or 100 nM of an mTOR inhibitor, Rapa or Torin1, for 6 days were subjected to Western blotting using antibodies (1:1000 dilutions) against LC3 (LC3-I; 16kDa, LC3-II; 13 kDa) and α-tubulin (55 kDa). The experiments were performed in triplicate using fresh preparations.

**Figure 2 biomedicines-12-02604-f002:**
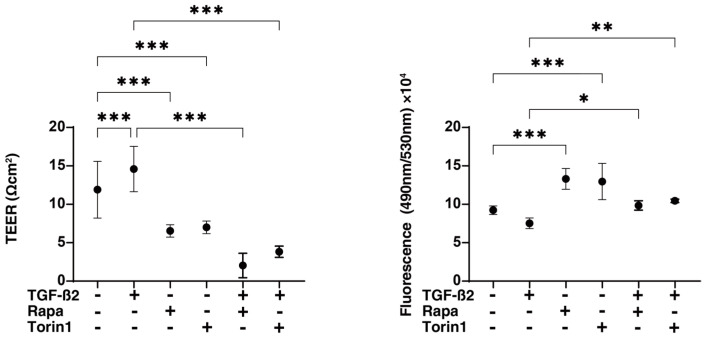
Effects of mTOR inhibitors on the barrier capacity of HTM 2D monolayers. For analysis of the barrier capacity of HTM 2D monolayers, TEER and FITC dextran permeability measurements were carried out under several conditions: untreated control and treated with 5 ng/mL TGF-β2 and/or 100 nM of an mTOR inhibitor, Rapa or Torin1. Plots of electric resistance (Ωcm^2^) by TEER and FITC dextran level are shown in panels A and B, respectively. The experiments were repeated three times using fresh preparations (n = 5 each). * *p* < 0.05; ** *p* < 0.01; *** *p* < 0.005.

**Figure 3 biomedicines-12-02604-f003:**
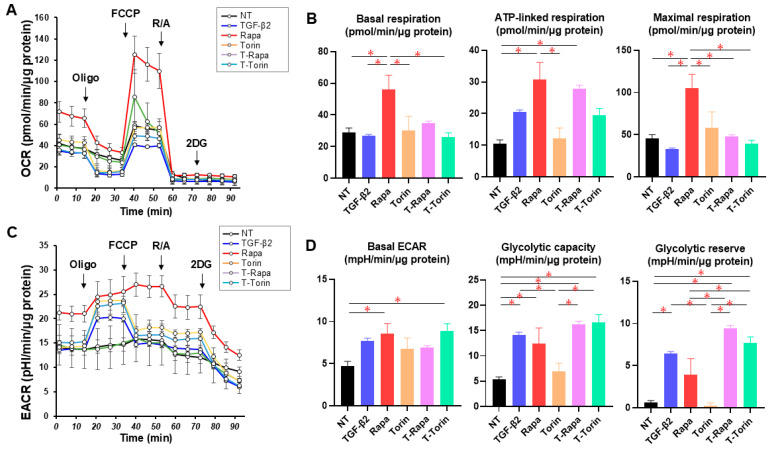
Effects of mTOR inhibitors on the cellular metabolic aspects of 2D HTM cells. In addition to the non-treated control (NT) cells, the 2D HTM cells were mono-treated with TGF-β2 or an mTOR inhibitor (Rapa or Torin) or treated with both TGF-β2 and an mTOR inhibitor (T-Rapa or T-Torin). Those samples were subjected to Seahorse real-time metabolic function analysis using an XFe96 Extracellular Flux Analyzer (n = 6). Plots of OCR (oxygen consumption rate) values (**A**) and key parameters of mitochondrial function: basal respiration (OCR at baseline—OCR with R/A), ATP-linked respiration (OCR at the baseline—OCR with Oligo), and maximal respiration (OCR with FCCP—OCR with R/A) (**B**), plot of ECAR (extracellular acidification rate) values (**C**), and key parameters of glycolytic functions: basal ECAR (ECAR at the baseline—ECAR with 2DG), glycolytic capacity (ECAR with Oligo—ECAR with 2DG), and glycolytic reserve (ECAR with Oligo –ECAR at baseline) (**D**). Oligo: oligomycin, FCCP: carbonyl cyanide p-trifluoromethoxyphenylhydrazone (FCCP), R/A: rotenone/antimycin A, 2DG: 2-deoxyglucose. * *p* < 0.05.

**Figure 4 biomedicines-12-02604-f004:**
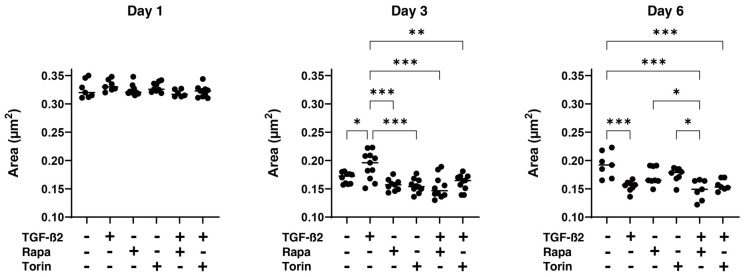
Effects of mTOR inhibitors on the mean sizes of the HTM spheroids. Under several conditions including the untreated control and cells treated with 5 ng/mL TGF-β2 and/or 100 nM of an mTOR inhibitor, Rapa or Torin1, the mean sizes (μm) of the spheroids were measured in triplicate using fresh preparations (total n = 16). The values at Days 1, 3, and 6 were plotted. * *p* < 0.05; ** *p* < 0.01; *** *p* < 0.005.

**Figure 5 biomedicines-12-02604-f005:**
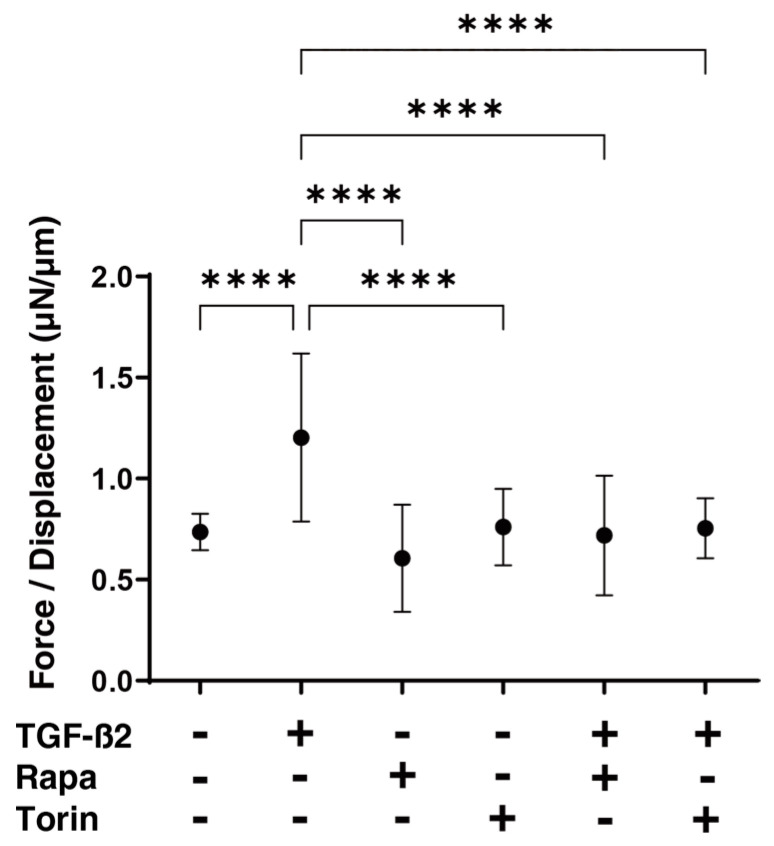
Effects of mTOR inhibitors on the stiffness of HTM spheroids. Under several conditions including the untreated control and cells treated with 5 ng/mL TGF-β2 and/or 100 nM of an mTOR inhibitor, Rapa or Torin1, the force (μN) required to produce 50% down-sizing in diameter during a period of 20 s of HTM spheroids was measured in triplicate using fresh preparations (total n = 16). The stiffness index values (μN/μm) of HTM spheroids were plotted. **** *p* < 0.001.

**Figure 6 biomedicines-12-02604-f006:**
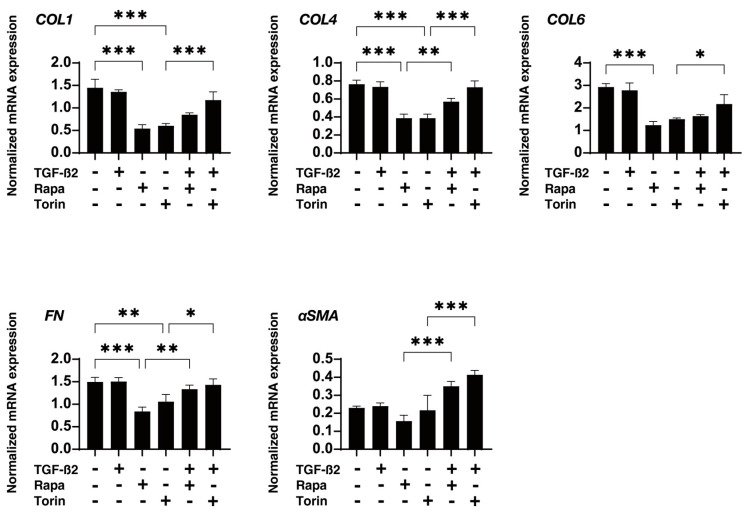
Effects of mTOR inhibitors on the mRNA expression of ECM molecules. Under several conditions including the untreated control and cells treated with 5 ng/mL TGF-β2 and/or 100 nM of an mTOR inhibitor, the mRNA expression of *COL1*, *COL4*, *COL6*, *FN*, and *aSMA* determined by qPCR in duplicate using 15 freshly prepared spheroids was plotted. * *p* < 0.05; ** *p* < 0.01; *** *p* < 0.005.

## Data Availability

The data that support the findings of this study are available from the corresponding author upon reasonable legal or ethical request.

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
