# Peer review of "mTOR Inhibitors Modulate the Biological Nature of TGF-β2-Treated or -Untreated Human Trabecular Meshwork Cells in Different Manners"

_biomedicines, 2024, doi:10.3390/biomedicines12112604_

Round 1
Reviewer 1 Report
Comments and Suggestions for Authors
1. The abstract should not contain any abbreviations.
2. The methodology is unclear in the Materials and Methods section, particularly in part 2.1, which addresses the preparation of two-dimensional (2D) and three-dimensional (3D) cultured glaucomatous human trabecular meshwork (HTM) cells. Additionally, the examination of drug-induced effects of mTOR inhibitors, such as 100 nM rapamycin (Rapa) or Torin1, needs clarification.
3. What parameters were used to measure when spheres reached 50% deformation?
4. The origin of equipment should be consistent and should include all items, specifying the company name, city, and country.
5. There is no mention of the origin for real-time PCR.
6. In the discussion section, there is no interpretation of the relationship between the effects of mTOR inhibitors on physical properties and the method of analysis.
7. The conclusion is missing.
8. The references need to be updated.
Comments on the Quality of English Language
Grammar and punctuation should be thoroughly revised.
Author Response
Dear Editor,
Thank you very much for the constructive comments concerning our manuscript “mTOR inhibitors modulate biological natures of TGF-b2 treated or untreated human trabecular meshwork cells in different manner”. We carefully checked all of the Editor and reviewers’ comments and prepared a revised version of our paper that takes these comments into account. The changes are listed below (changes according to reviewers’ comments: highlighted by yellow).
Editor comments
- As suggested, similarity rate should be reduced and changes are highlighted by blue.
- As suggested, self-citation is reduced.
Reviewer 1 comments
- The abstract should not contain any abbreviations.
Answer; We sincerely appreciate your excellent comment. As suggested, full spells of all abbreviations are included.
- The methodology is unclear in the Materials and Methods section, particularly in part 2.1, which addresses the preparation of two-dimensional (2D) and three-dimensional (3D) cultured glaucomatous human trabecular meshwork (HTM) cells. Additionally, the examination of drug-induced effects of mTOR inhibitors, such as 100 nM rapamycin (Rapa) or Torin1, needs clarification.
Answer; We sincerely appreciate your excellent comment. As suggested, details of 2D culture method are included and sentence related to drug induced effects of mTOR inhibitors are changed: ‘All studies were performed according with the tenets of the Declaration of Helsinki and were approved by the internal review board of Sapporo Medical University. As in vitro glaucomatous HTM models, immortalized HTM cells (Applied Biological Materials Inc., Richmond, Canada) [27] were 2D and 3D cultured in the absence and presence of 5 ng/mL of TGF-β2 as reported previously [28]. Briefly, the HTM cells were maintained in 150 mm planar culture dishes at 37°C in high glucose Dulbecco's Modified Eagle Medium (HG-DMEM) medium containing 10% fetal bovine serum (FBS), 1% L-glutamine and 1% antibiotic-antimycotic, until reaching 90% confluence by changing the medium every other day. For spheroid generation, 2D cultured HTM cells were collected and resuspended in the same culture medium supplemented with 0.25% methylcellulose. Cell numbers were adjusted to approximately 20,000 cells in 28 ml of culture medium and placed in each well of a hanging droplet 3D culture plate (# HDP1385, Sigma-Aldrich. St. Louis, MO, U.S.A). Thereafter, spheroid culture was carried out for a 6-day period with daily change of half of the medium with a fresh medium.
To evaluate the drug-induced effects of mTOR inhibitors on 5 ng/ml TGF-β2-treated or untreated 2D and HTM spheroids, 100 nM rapamycin (Rapa) or Torin1 was administered from Day 1 to Day 6 of the culture period. The concentrations of TGF-b2 [28,29] and mTOR inhibitors were confirmed to be the optimum concentrations based on data previously reported [30,31].’.
- What parameters were used to measure when spheres reached 50% deformation?
Answer; We sincerely appreciate your excellent comment. As pointed out, those are corrected to ‘To obtain an index, ratio of force/displacement (μN/μm) expressing stiffness, the force required for the diameter of a single living spheroid to reach 50% of its length (μm) was measured by applying a micro-compressor system (MicroSquisher, CellScale, Waterloo, ON, Canada).’ to avoid ambiguity.
- The origin of equipment should be consistent and should include all items, specifying the company name, city, and country.
Answer; We sincerely appreciate your excellent comment. As suggested, those are unified.
- There is no mention of the origin for real-time PCR.
Answer; We sincerely appreciate your excellent comment. As suggested, the origin for real-time PCR is included.
- In the discussion section, there is no interpretation of the relationship between the effects of mTOR inhibitors on physical properties and the method of analysis.
Answer; We sincerely appreciate your excellent comment. As suggested, discussion is rewritten to include interpretation of the relationship between the effects of mTOR inhibitors on physical properties and the method of analysis.
- The conclusion is missing. As suggested, conclusion is included: ‘In conclusion, taken together with previous observations that mTOR inhibitors ameliorated TGF-b2-induced fibrogenesis [59][60], our collective results suggested that mTOR inhibitors may have beneficial effects on 1) cell size and number of cells and 2) autophagy of TGF-b2-untreated (healthy) HTM and TGF-b2-treated (glaucomatous) HTM, and 3) cellular metabolic functions of TGF-b2-untreated (healthy) HTM. However, mTOR is involved in various biological functions and mTOR inhibitors thus have many pharmacological effects including regulation of autophagy. In addition, the effects of short-term exposure and long-term exposure to Rap have been shown to be different [12]. Therefore, additional investigations using various effectors related to mTOR signaling with primary cultured glaucomatous HTM cells will be required for a better understanding of the effects of mTOR inhibitors on glaucomatous and non-glaucomatous HTMs.’.
Answer; We sincerely appreciate your excellent comment.
- The references need to be updated.
Answer; We sincerely appreciate your excellent comment. As suggested, the references to be updated as possible as I can.
- Comments on the Quality of English Language: Grammar and punctuation should be thoroughly revised.
Answer; We sincerely appreciate your excellent comment. As suggested, English is carefully checked by a native English-speaking scientist.
Reviewer 2 comments
- This manuscript describes how mTOR inhibitors modulate the biology of TGF-β2 treated and untreated human trabecular meshwork cells. The authors find differences, which they describe, although no significant conclusions are made. Overall, the research is sound and well-presented; however, it's significance is lacking. Essentially, the study is overly descriptive in nature. It's not clear why the reader should be interested and what conclusions the reader should draw.
Answer; We sincerely appreciate your excellent comment. As suggested, discussion is rewritten to be more easily understood conclusion of the present study by readers by including interpretation of the relationship between the effects of mTOR inhibitors on physical properties and the method of analysis.

Reviewer 2 Report
Comments and Suggestions for Authors
This manuscript describes how mTOR inhibitors modulate the biology of TGF-β2 treated and untreated human trabecular meshwork cells. The authors find differences, which they describe, although no significant conclusions are made. Overall, the research is sound and well-presented; however, it's significance is lacking. Essentially, the study is overly descriptive in nature. It's not clear why the reader should be interested and what conclusions the reader should draw.
Author Response

(The authors gave the same response as above.)

Round 2
Reviewer 1 Report
Comments and Suggestions for Authors
No more comments